# Changes in the Proteome in the Development of Chronic Human Papillomavirus Infection—A Prospective Study in HIV Positive and HIV Negative Rwandan Women

**DOI:** 10.3390/cancers13235983

**Published:** 2021-11-28

**Authors:** Emile Bienvenu, Marie Francoise Mukanyangezi, Stephen Rulisa, Anna Martner, Bengt Hasséus, Egor Vorontsov, Gunnar Tobin, Daniel Giglio

**Affiliations:** 1College of Medicine and Health Sciences, University of Rwanda, KK 737 St, Gikondo, Kigali, Rwanda; ebienvenu3@gmail.com (E.B.); francizi@yahoo.fr (M.F.M.); 2College of Medicine and Health Sciences, University of Rwanda, University Teaching Hospital of Kigali (UTHK), KN 4 Ave, Kigali, Rwanda; s.rulisa@gmail.com; 3TIMM Laboratory, Sahlgrenska Center for Cancer Research, Department of Infectious Diseases, Institute of Biomedicine, Sahlgrenska Academy, University of Gothenburg, 41390 Gothenburg, Sweden; anna.martner@microbio.gu.se; 4Department of Oral Medicine and Pathology, Institute of Odontology, Sahlgrenska Academy, University of Gothenburg, 40530 Gothenburg, Sweden; bengt.hasseus@odontologi.gu.se; 5Proteomics Core Facility, Sahlgrenska Academy, University of Gothenburg, 40530 Gothenburg, Sweden; egor.vorontsov@gu.se; 6Department of Pharmacology, Sahlgrenska Academy, University of Gothenburg, 40530 Gothenburg, Sweden; gunnar.tobin@pharm.gu.se; 7Department of Oncology, Sahlgrenska Academy, University of Gothenburg, Sahlgrenska University Hospital, 41345 Gothenburg, Sweden

**Keywords:** cervical cancer, intraepithelial lesion, HPV, HIV, proteomics, Rwanda

## Abstract

**Simple Summary:**

Cervical cancer is the predominant cause of female cancer deaths in Sub-Saharan Africa. Chronic infection of the uterine cervix by the human papillomavirus (HPV) is the most important factor for the development of cervical cancer. HPV screening may be used to screen out women at risk of cervical cancer; however, the majority of HPV infections will eventually heal, and at present, there are no screening methods to detect which HPV infections that will become chronic. In the present study, we followed HIV-negative and HIV-positive Rwandan women with cervical HPV infections for two years with repeated samplings from the uterine cervix. We identified protein biomarkers that correlate with the potential risk for women to develop cervical cancer. By including the identified biomarkers in cervical screening programs, we are able, potentially, to identify those women with cervical HPV infections, who should be carefully monitored in the future.

**Abstract:**

Background: Effects on the proteome when a high risk (HR)-HPV infection occurs, when it is cleared and when it becomes chronic were investigated. Moreover, biomarker panels that could identify cervical risk lesions were assessed. Methods: Cytology, HPV screening and proteomics were performed on cervical samples from Rwandan HIV+ and HIV- women at baseline, at 9 months, at 18 months and at 24 months. Biological pathways were identified using the String database. Results: The most significantly affected pathway when an incident HR-HPV infection occurred was neutrophil degranulation, and vesicle-mediated transport was the most significantly affected pathway when an HR-HPV infection was cleared; protein insertion into membrane in chronic HR-HPV lesions and in lesions where HR-HPVs were cleared were compared; and cellular catabolic process in high-grade lesions was compared to that in negative lesions. A four-biomarker panel (EIF1; BLOC1S5; LIMCH1; SGTA) was identified, which was able to distinguish chronic HR-HPV lesions from cleared HR-HPV/negative lesions (sensitivity 100% and specificity 91%). Another four-biomarker panel (ERH; IGKV2-30; TMEM97; DNAJA4) was identified, which was able to distinguish high-grade lesions from low-grade/negative lesions (sensitivity 100% and specificity 81%). Conclusions: We have identified the biological pathways triggered in HR-HPV infection, when HR-HPV becomes chronic and when cervical risk lesions develop. Moreover, we have identified potential biomarkers that may help to identify women with cervical risk lesions.

## 1. Introduction

Cervical cancer is the most important cause of cancer death among women living in sub-Saharan Africa [1]. Chronic infection of the uterine cervix by human papillomavirus (HPV) constitutes the most important risk factor for the development of cervical cancer [2]. In many countries in sub-Saharan Africa, cervical infections by high-risk HPV (HR-HPV) types are common. While HPV screening has been implemented in many high- and middle-income countries, HPV screening is difficult to implement in most countries in sub-Saharan Africa due to limited health care resources. HIV is prevalent in sub-Saharan Africa, and it is a known risk factor for contracting HPV and for developing chronic cervical HPV infections and cervical intraepithelial neoplasia (CIN) [3]. In our previous study, we showed that, among HIV positive Rwandan women attending the HIV clinic at the University Teaching Hospital in Kigali, Rwanda, 56% of them presented with chronic cervical HR-HPV infections [3]. The high percentage of HR-HPV positivity among well-monitored and well-treated HIV positive Rwandan women was also shown by Murenzi et al. [4].

Different approaches have been tested to reduce the number of cases of cervical cancer in developing countries. Cytology as a screening method is difficult to implement in developing countries due to a limitation in the number of cytologists. We and other research groups have shown that HPV screening may be used to identify women with high-grade cervical lesions in Rwanda [5,6]. While different approaches may be undertaken to reduce the number of cervical cancer cases, there is a risk of over-treatment, and the follow-up of patients at risk of cervical cancer is difficult to organize in many sub-Saharan countries. 

Proteomics has been used for identifying biomarkers correlating with the presence of premalignant lesions and cancer of the uterine cervix [7,8]. Biomarkers may not only be identified directly in the uterine cervix but also in the cervicovaginal fluid [9,10], which makes self-sampling possible for screening. For example, a 7-protein signature from cervicovaginal fluid samples, which discriminated between CIN2+ lesions and controls, was identified by Gutiérrez et al. [10].

The present study was conducted in HIV positive and HIV negative women in Rwanda, where women were sampled from the uterine cervix at repeated occasions during a two-year period. The study had two general aims. First, we wanted to assess what happens in the uterine cervix when a cervical HR-HPV infection occurs and is eventually eradicated, when an HR-HPV infection becomes chronic and when high-grade intraepithelial lesions (HSIL = CIN2+/CIN3+) and cervical cancer are developed. Secondly, we wanted to identify biomarker panels that could discriminate between high-grade and chronic HR-HPV positive lesions and benign cervical lesions.

## 2. Materials and Methods

### 2.1. Description of the Cohort

The description of the present cohort has been presented in detail in our previous study [3]. In brief, HIV-positive Rwandan women were recruited at the HIV clinic at the University Teaching Hospital of Kigali (CHUK), and HIV-negative women were recruited at the outpatient gynaecology and obstetrics clinics at CHUK and at the University Teaching Hospital of Butare (CHUB) between 2015 and 2017. A sample with an endocervical brush (Hologic Inc.; Marlborough, MA, USA) was taken from the uterine cervix, and the brush was rinsed in PreservCyt transport medium; cytology and HPV screening were then performed (see Appendix A). A second sample was taken with a brush from the cervix for proteome analyses and stored in RNAlater^®^ (Thermo Fisher Scientific, Waltham, MA, USA). The initial intent was to do both transcriptomics and proteomics; however, the concentrations of RNA in the samples were too low to do transcriptomics. Participating women underwent the same sampling procedure as described above after 9, 18 and 24 months. Possible high-risk (HR)-HPVs were grouped together with HR-HPVs. The women categorized as having an HR-HPV chronic infection were those presenting with HR-HPV infections at baseline that were still present at follow-up after 24 months (*n* = 14). Women categorized as HR-HPV cleared were, at first sampling, HR-HPV positive and, at follow-up, were HPV negative at two sampling occasions (*n* = 9). Women categorized as HR-HPV incident were, at first sampling, HPV negative and, at the next follow-up, HR-HPV positive (*n* = 7). Average time span from baseline sampling to follow-up sampling for HR-HPV cleared and HR-HPV incident groups was 8.1 ± 0.4 months (*n* = 7 + 9). HPV negative women were defined as women who were HPV negative at all sampling occasions and with no cytological changes (0–24 months; negative group; *n* = 14). The HR-HPV chronic group was subdivided into high-grade and low-grade lesions (Table 1). Samples from the HPV chronic group where cytology was graded as HSIL (*n* = 5), adenocarcinoma in situ (AIS; *n* = 1) or squamous cell carcinoma (SCC; *n* = 1) were defined as high-grade lesions, and, where cytology was graded as low-grade squamous intraepithelial lesion (LSIL; *n* = 2), atypical squamous cells, undetermined significance (ASCUS; *n* = 3) or normal cytology (*n* = 2), they were grouped and defined as low-grade lesions (table). Age, HIV status, cytology and HPV status and HPV types are described in the table. 

First, we wanted to assess whether the proteome changed in response to the development of the HR-HPV infection and what happens in the proteome when the HR-HPV infection is cleared (group A and group B, respectively; Table 1). Here, the proteome across paired consecutive sampling occasions was compared. Samples were compared between either the first or second sampling occasion and the first of two negative sampling occasions. Second, we wanted to assess whether the proteome at baseline was different between women clearing their HR-HPV infection, women developing chronic HR-HPV infections and HR-HPV negative women (group B, group C and group D, respectively; Table 1). Since no HIV negative women developed chronic HR-HPV infections in our cohort [3], the chronic HR-HPV group consisted solely of HIV positive women. The HPV cleared group consisted of six HIV positive (67%) and three HIV negative women (33%). Third, we wanted to assess whether the proteome changed according to cytology status by comparing high-grade lesions (part of group C), low-grade lesions (another part of group C) and negative lesions (constituting the same group as HPV negative group = group D; table). 

### 2.2. Sample Preparation for Proteomic Analysis, LC-MS/MS Analysis and Data Analysis

The methods used for sample preparation for proteomic analysis, Liquid chromatography–Mass Spectrometry (LC-MS/MS) analysis and proteomic data analysis, are described in Appendix A.

### 2.3. Statistics, Venn Diagrams and Volcano Plots

The statistical analyses and software used for statistics as well as Venn diagrams and volcano plots are described in Appendix A.

### 2.4. Cluster Analysis

Squared Euclidean distance-based hierarchical clustering analyses were performed with Perseus software (v1.6.14.0) for either all changed proteins between groups or the 40 most changed proteins according to the absolute fold-change [11].

### 2.5. Protein-Protein Network Analysis

The Search Tool for the Retrieval of Interacting Genes/Proteins (String) software (version 11.0) was used for protein-protein network analysis [12]. The confidence was set to medium (0.400) for the minimum interaction score. 

### 2.6. Ethical Statement

The Institutional Review Board at the College of Medicine and Health Sciences, University of Rwanda and the Regional Ethical Review Board in Gothenburg approved the present study. Written informed consent was provided by each participant in the study. The code is 642-15 for the Swedish approval and 081/2015 for the Rwandan approval

## 3. Results

### 3.1. Proteome Changes in the Incident and Cleared HR-HPV Groups

In total, 5939 proteins were identified in the 44 cervical samples. First, we assessed whether the proteome changed in samples that were HPV negative at baseline and where HR-HPV infections were present at follow-up (incident HR-HPV group; *n* = 7). In paired samples between uninfected at baseline and HR-HPV infected at follow-up, 181 proteins were up-regulated, and 133 proteins were down-regulated at follow-up (HPV incident group; Figure 1a). Close to all of the 40 most changed proteins in the HPV incident group were up-regulated (Figure 1b).

Protein-protein network analysis of the 40 most changed proteins in the HR-HPV incident group showed that 27 of these genes were related to neutrophil degranulation (false discovery rate; FDR = 1.12 × 10^−30^); Figure 2a). Moreover, when all proteins with changed expression following incident HR-HPV infection were included, neutrophil degranulation was the biological pathway that most strongly associated with incident infection, followed by vesicle-mediated transport, leukocyte mediated immunity, myeloid leukocyte activation and leukocyte activation involved in immune response (Appendix A).

Next, we assessed whether the proteome of the uterine cervix changed in response to clearance of HR-HPV infection. In paired samples between HR-HPV infected at baseline and uninfected at follow-up, 71 proteins were up-regulated, and 34 proteins were down-regulated at follow-up (HR-HPV cleared group; *n* = 9; Figure 1a). The 40 most changed proteins are displayed in Figure 1b. The biological processes that differed the most before and after clearance of the HR-HPV infection were vesicle-mediated transport, translational initiation, establishment of protein localization, intracellular transport and establishment of localization in cell (Figure 2b). Proteins that changed from baseline in the opposite direction in the incident and the cleared HR-HPV groups were: far upstream element-binding protein 3 (FUBP3), charged multivesicular body protein 4a (CHMP4A), eukaryotic translation initiation factor 2 subunit 2 (eIF2), EH domain-binding protein 1 (EHBP1), serpin B8, aminoacyl tRNA synthase complex-interacting multifunctional protein 1 (AMP1), polyadenylate-binding protein 4, epidermal growth factor receptor substrate 15-like 1 (EPS15L1) and probable bifunctional dTTP/UTP pyrophosphatase/methyltransferase protein (ASMTL; Figure 1c). These proteins were all down-regulated in the incident HR-HPV group and up-regulated in the cleared HR-HPV group. In contrast, serpin B8 was up-regulated and FUBP3 was down-regulated in both groups from baseline.

### 3.2. Proteome Changes in the Chronic and Cleared HR-HPV Groups

In the chronic HR-HPV group, 57 proteins were significantly more expressed and 61 proteins less expressed, as compared with the negative group, and 13 proteins were significantly more expressed and 14 proteins less expressed, as compared with the cleared HR-HPV group (Figure 3a,b). Seven proteins were uniquely differently expressed in the HPV chronic group, as compared with the HPV cleared group and the negative group, i.e., LIM and calponin homology domains-containing protein 1 (LIMCH1), small glutamine-rich tetratricopeptide repeat-containing protein alpha (SGTA), biogenesis of lysosome-related organelles complex 1 subunit 5 (BLOC1S5), eukaryotic translation initiation factor 1 (EIF1), grpE protein homolog 1, mitochondrial (GRPEL1), optineurin (OPTN) and ezrin (EZR; Figure 3c). LIMCH1 was the protein that was mostly separated in the volcano plot. The largest AUC of the ROC curves for the seven biomarkers was for EIF1 with an AUC of 0.89 (95% CI: 0.77–1.00; Figure 3d). With a cut-off value for EIF1 equal to or more than 0.258 (log2 expression), the sensitivity was 93%, and the specificity was 83% to detect chronic HR-HPV lesions among cleared HR-HPV lesions and negative lesions. 

### 3.3. Biomarker Panels Identifying Chronic HR-HPV Lesions

The following equations combining 2–4 biomarkers were created:Two-biomarker combined predictor = 3.503 × EIF1 + 1.546 × BLOC1S5 − 1.250

With a cut-off value for the three-biomarker combined predictor greater than or equal to −0.2749, the sensitivity was 86%, and the specificity was 91% to detect chronic HR-HPV lesions among cleared HR-HPV and negative lesions. The AUC was 0.92 (95% CI: 0.82–1.00; Figure 3d).
Three-biomarker combined predictor = 1.731 × BLOC1S5 − 2.439 × LIMCH1 + 2.714 × EIF1 − 0.779

With a cut-off value for the three-biomarker combined predictor greater than or equal to −0.4934, the sensitivity was 93%, and the specificity was 91% to detect chronic HR-HPV lesions among cleared HR-HPV and negative lesions. The AUC was 0.95 (95% CI: 0.89–1.00; Figure 3d).
Four-biomarker combined predictor = 12.982 × EIF1 + 5.907 × BLOC1S5 − 10.489 × LIMCH1 − 21.989 × SGTA − 13.160

With a cut-off value for the four-biomarker combined predictor greater than or equal to −1.3580, the sensitivity was 100%, and the specificity was 91% to detect chronic HR-HPV lesions among cleared HR-HPV and negative lesions. The AUC was 0.99 (95% CI: 0.96–1.00; Figure 3d).

### 3.4. Biological Processes in the Chronic HR-HPV, Cleared HR-HPV and HPV Negative Groups

The biological processes with the lowest FDRs between the cleared HR-HPV group and chronic HR-HPV group were protein insertion into the membrane, protein targeting to mitochondrion, cellular component organization, protein localization to organelle, establishment of localization and cellular localization (Figure 4a,b). The biological processes with the lowest FDRs between the cleared HR-HPV group and the HPV negative group were the cofactor metabolic process and the organonitrogen compound metabolic process (Figure 4c,d). The biological processes with the lowest FDRs between the chronic HR-HPV group and the HPV negative group were the cellular catabolic process, the organonitrogen compound metabolic process, protein deubiquitination, the cofactor metabolic process, proteolysis involved in the cellular protein catabolic process and the organic substance catabolic process (Appendix A). 

### 3.5. Proteome Changes in the Low-Grade and High-Grade Lesion Groups

In the high-grade lesion group (*n* = 7), 133 proteins were significantly more expressed and 57 proteins significantly less expressed, as compared with the negative group (*n* = 14), and 91 proteins were significantly more expressed, and 24 proteins significantly less expressed, as compared with the low-grade lesion group (*n* = 7; Figure 3a and Figure 5a). Seven potential biomarkers were identified in the volcano plot where the high-grade group was significantly different from the low-grade and negative groups, i.e., immunoglobulin kappa variable 2–30 (IGKV2-30), enhancer of rudimentary homolog (ERH), sigma intracellular receptor 2 (TMEM97), SWI/SNF-related matrix-associated actin-dependent regulator of chromatin subfamily B member 1 (SMARCB1), DnaJ homolog subfamily A member 4 (DNAJA4), double-strand-break repair protein rad21 homolog (RAD21) and heterogeneous nuclear ribonucleoprotein A1 (HNRNPA1; Figure 5b). With a cut-off value for IGKV2-30 equal to or less than −0.588 (log2 expression), the sensitivity was 100%, and the specificity was 76% to detect high-grade lesions among low-grade and negative lesions (Figure 5c). AUC was 0.94 (95% CI: 0.85–1.00).

### 3.6. Biomarker Panels Identifying High-Grade Lesions

The following equations combining 2–4 biomarkers were created:Two-biomarker combined predictor = −3.570 × DNAJA4 − 1.504 × IGKV2-30 − 3.083

With a cut-off value for the three-biomarker combined predictor greater than or equal to −2.2984, the sensitivity was 100%, and the specificity was 76% to detect high-grade lesions among low-grade and negative lesions. AUC was 0.95 (95% CI: 0.86–1.00; Figure 5c).
Three-biomarker combined predictor = 1.944 × TMEM97 − 0.979 × DNAJA4 − 2.168 × IGKV2-30 − 3.665

With a cut-off value for the three-biomarker combined predictor greater than or equal to −2.2166, the sensitivity was 100%, and the specificity was 81% to detect high-grade lesions among low-grade and negative lesions. AUC was 0.97 (95% CI: 0.91–1.00; Figure 5c).
Four-biomarker combined predictor = 0.242 × ERH − 2.045 × IGKV2-30 + 1.786 × TMEM97 − 1.098 × DNAJA4 − 3.546

With a cut-off value for the four-biomarker combined predictor greater than or equal to −2.3263, the sensitivity was 100%, and the specificity was 81% to detect high-grade lesions among low-grade and negative lesions. AUC was 0.97 (95% CI: 0.90–1.00; Figure 5c). Among biomarkers identified for the chronic HR-HPV group, EIF1 was also differently expressed in the high-grade, as compared with the negative lesions, and in the low-grade, as compared with the negative lesions.

### 3.7. Networks in High-Grade and Low-Grade Cervical Lesions

The biological processes with the lowest FDRs between high-grade lesions and low-grade lesions were mRNA splicing, via spliceosome, mRNA processing, RNA splicing, the mRNA metabolic process, the nucleobase-containing compound metabolic process and RNA processing (Figure 6a,b). For proteins differently expressed between low-grade lesions and negative lesions, the protein network did not have significantly more interactions than expected (Figure 6c,d). Negative regulation of protein folding was, however, identified as a significant biological pathway (Figure 6c). The biological processes with the lowest FDRs between high-grade lesions and negative lesions were the cellular catabolic process, the organonitrogen compound metabolic process, protein deubiquitination, the cofactor metabolic process, proteolysis involved in the cellular protein catabolic process and the organic substance catabolic process (Figure 7a,b).

## 4. Discussion

### 4.1. The Immune System Is Involved in HPV Infection but Not in Clearance of Infection

The present study showed that immunological processes take place in the uterine cervix in response to an HR-HPV infection. The biological pathway that was most significantly affected following incident HPV infection was neutrophil degranulation. Studies show that neutrophil degranulation is enhanced in HPV-induced dermal warts, and it is suggested that neutrophils are involved in cervical carcinogenesis by suppressing T-cell activity [13,14,15]. In contrast, the involvement of the immune system in the clearance of HPV infection did not turn out to be a biological pathway of major importance. Instead, vesicle-mediated transport, translational initiation, the establishment of protein localization, intracellular transport and the establishment of localization in cell were the biological pathways activated in the HPV cleared group. Our results indicate that clearance of HR-HPV infection may involve changes in the complex processes required for the establishment of HPV infection [16]. 

We found that CHMP4A, eIF2, EHBP1, AIMP1, PABPC, EPS15L1 and ASMTL were down-regulated in the incident HR-HPV infection and up-regulated in the cleared HR-HPV infection. The heterotrimer eIF2 consists of α (subunit 1), β (subunit 2) and γ (subunit 3). Studies show that eIF2 counteracts viral infections, and phosphorylation of eIF2α leads to the inhibition of HPV18 E6 protein synthesis [17]. The eukaryotic translation initiation factors (EIFs) seem to be of importance in HPV infection and in the pathogenesis of different cancer forms including cervical cancer [18,19,20,21,22,23]. Besides eIF2α, EHBP1 is of particular interest among our identified proteins. This protein may bind to Rab8 and regulate apical exocytic protein transport in epithelial cells [24]. Rab proteins regulate the endosomal transport of viruses including dengue and West Nile Virus [25,26]. Rab-GTPase also regulates the entry and transport of HPV to the nucleus [27]. AIMP1 may regulate antiviral immunity and polarizes immunity towards Th1 polarization [28,29]. 

### 4.2. Proteins Involved in the Development of HR-HPV Chronicity

Few biological pathways could be identified changing in the cleared HR-HPV group vs. the chronic HR-HPV group. Among the biological pathways identified, pathways regulating how proteins were established in membranes were affected. There was also a limited number of biological pathways changed between the cleared HR-HPV group and the negative group. The time span between the paired samplings was eight months, and the results may indicate that the establishment of HPV infection takes place during a longer period of time than the period of time for clearance of HPV infection. The biological pathways that changed in response to HR-HPV chronicity were regulating metabolism. The organonitrogen compound metabolic process had low FDRs both in comparisons between the cleared HR-HPV group and the negative controls as well as between the chronic HR-HPV group and negative controls. 

### 4.3. Biomarkers Identifying Patients with Chronic HR-HPV Cervical Infections

LIMCH1, SGTA, BLOC1S5, EIF1, GRPEL1, optineurin and ezrin were proteins identified as potential biomarkers to screen out HR-HPV lesions. LIMCH1 was recently identified as a biomarker for aggressive cervical cancer [30]. The same study showed that EIF5A2 was up-regulated in aggressive cervical cancer, a protein that interacts with the HPV16 E6 [19]. In our panel of biomarkers, eIF1 was down-regulated in chronic HR-HPV lesions compared with cleared HR-HPV lesions and negative controls. eIF1 was differently expressed between high-grade lesions and controls and between low-grade lesions and controls. EIF4A2 was also differently expressed in high-grade lesions compared with low-grade lesions and controls, and EIF4B was differently expressed between high-grade cervical lesions and controls. High ezrin expression is correlated with poor prognosis in cervical cancer, and knockdown of ezrin inhibits migration and invasion in cervical cancer cells [31,32]. Moreover, ezrin expression seems to be a prognostic marker for the progression of cervical premalignant lesions to cervical cancer [33]. Optineurin is an autophagy adaptor protein and is a marker for poor prognosis in hepatocellular carcinoma [34]. Among identified biomarkers, small glutamine-rich tetratricopeptide repeat protein (SGT) and ezrin have been shown to interact with HIV; i.e., SGTA has been shown to interact with Vpu and Gag of HIV-1 and may inhibit HIV-1 particle release, and ezrin may take part in the replication of HIV [35,36]. HIV status did not, however, seem to be a bias explaining the expressional differences for ezrin and SGTA between the chronic HR-HPV group and cleared HR-HPV lesions/negative controls (data not shown).

### 4.4. RNA Processing in High-Grade Lesions

We found evidence that pathways regulating RNA were the most important pathways for distinguishing high-grade lesions from other cervical samples. Alternative splicing is important for the life cycle of HPV [37]. Our findings are in line with the report by Pottier et al. showing with proteomics that DNA replication and RNA splicing are enhanced in HSIL [38]. 

### 4.5. Biomarker Panels Identifying High-Grade Lesions

The two-four protein biomarker panel gave a high accuracy in predicting that the sample was a high-grade lesion among other lesions. Combining DNAJA4 with IGKV2-30 gave an AUC of 0.95 of the ROC curve. Also including ERH and TMEM97 in the biomarker panel only increased the AUC to 0.97. While the proteins in the biomarker panel have not been identified in the pathogenesis of cervical dysplasia and cancer, the proteins are identified as regulators of dysplasia and cancer in other tissues. The heat shock protein (HSP) family member DNAJA4 is induced in T-cells in heat shock and in HIV-1 infection [39]. The V region is the variable domain of the light chains of the immunoglobulin, which participates in antigen recognition [40]. Cell line studies show that ERH may regulate proliferation in ovarian cancer and melanoma and regulates migration and invasion in urinary bladder cancer [41,42,43]. TMEM97 also regulates cell proliferation in cancer [44].

The weakness of the study is that the number of women assessed was low, and for the chronic HR-HPV group, we only assessed HIV positive women. Due to a limited number of chronic HR-HPV samples, we could not group according to cytological categories, and therefore, one limitation is that the HR-HPV chronic group constitutes a mixed group for cytology. Studies show that HIV positive women have more CD8+ lymphocytes in HPV-positive cervical lesions, compared with HIV negative women [45,46]. Therefore, we cannot rule out the possibility that HIV infection may be a bias in the analyses. Moreover, our findings from cytology were not confirmed with histology, and consequently, cytology—and not histology—was used as the gold standard for the ROC curves.

## 5. Conclusions

We have identified biomarker panels that may potentially distinguish high-risk cervical lesions from low-risk lesions. Moreover, we found evidence for the involvement of neutrophils in the development of HR-HPV infection. The panels of biomarkers are currently validated in a new cohort of HIV-positive women in Rwanda and will also be validated in a cohort of HIV-positive women in Ethiopia.

## Figures and Tables

**Figure 1 cancers-13-05983-f001:**
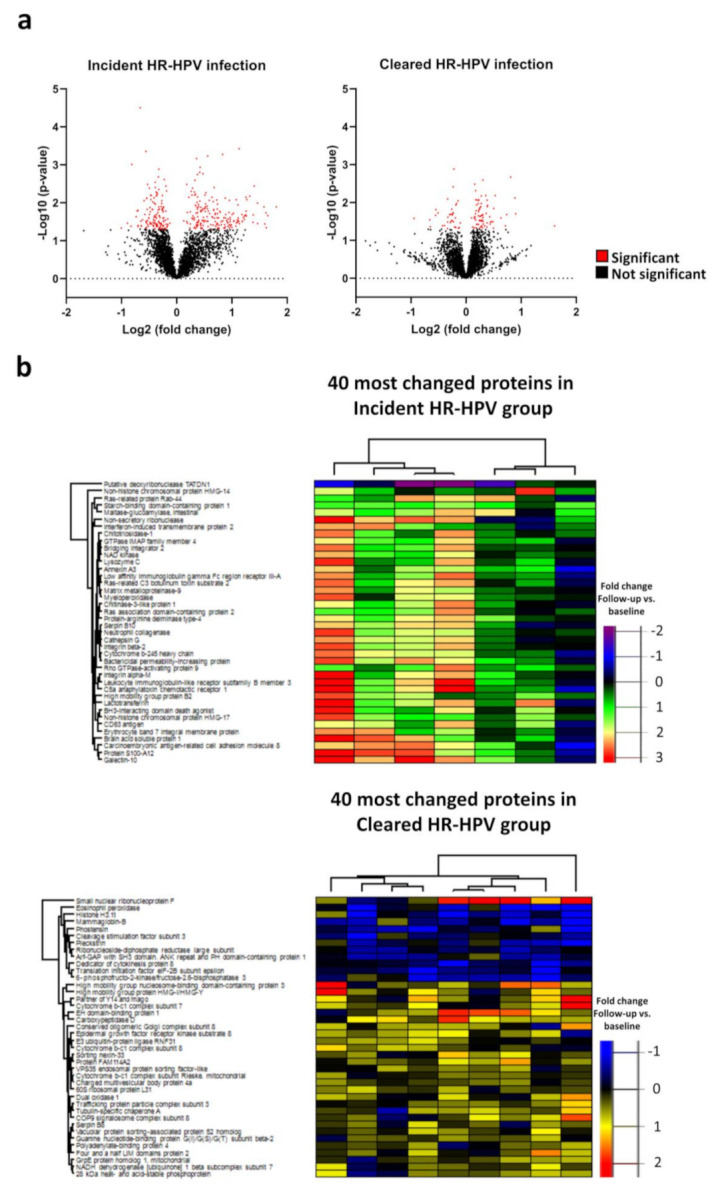
Volcano plots of log2 fold change vs. log10 *p*-value of changed proteins from baseline to follow-up in the incident and cleared HR-HPV groups, respectively. Red dots show significantly changed proteins and black dots show non-significantly changed proteins (**a**); heatmaps of the 40 most changed proteins in the incident and cleared HR-HPV groups, respectively (**b**); Venn diagram of incident and cleared HR-HPV groups and the 9 proteins changed in both groups (**c**).

**Figure 2 cancers-13-05983-f002:**
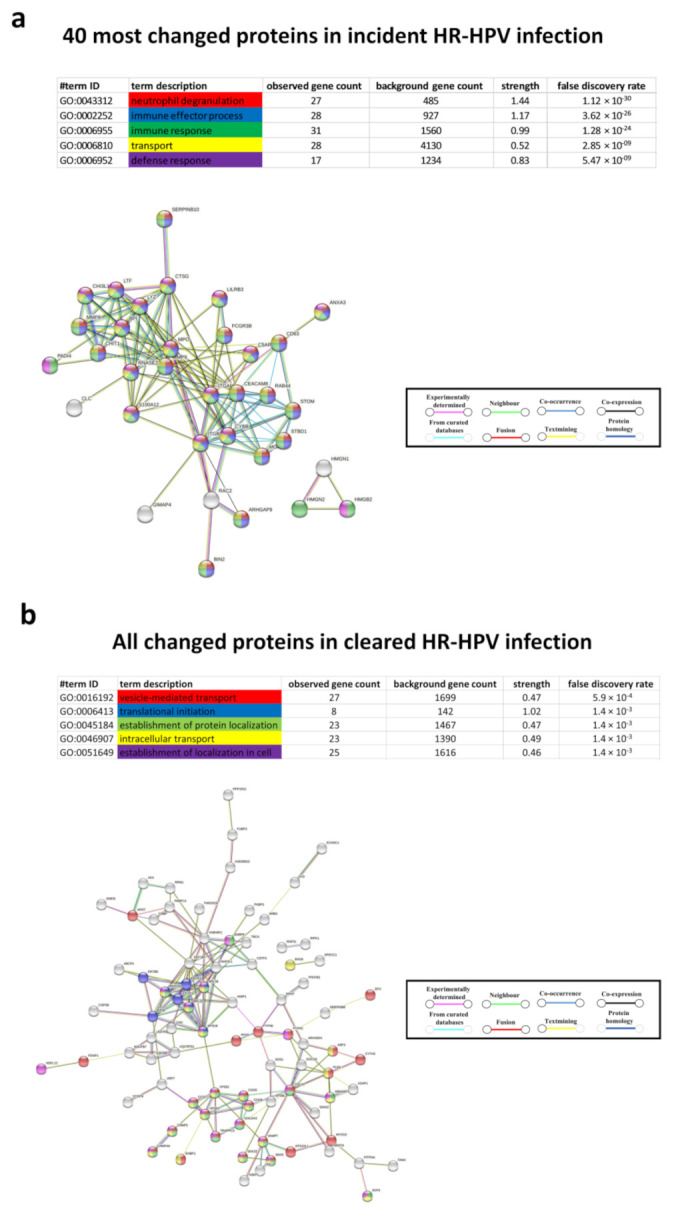
Protein-protein network analysis of the biological processes of the 40 most changed proteins in the incident HR-HPV group (**a**) and of all changed proteins in the cleared HR-HPV group (**b**). The biological processes with the lowest FDRs are indicated by colour, and interactions are indicated by colour and line type. Disconnected nodes are hidden in the network.

**Figure 3 cancers-13-05983-f003:**
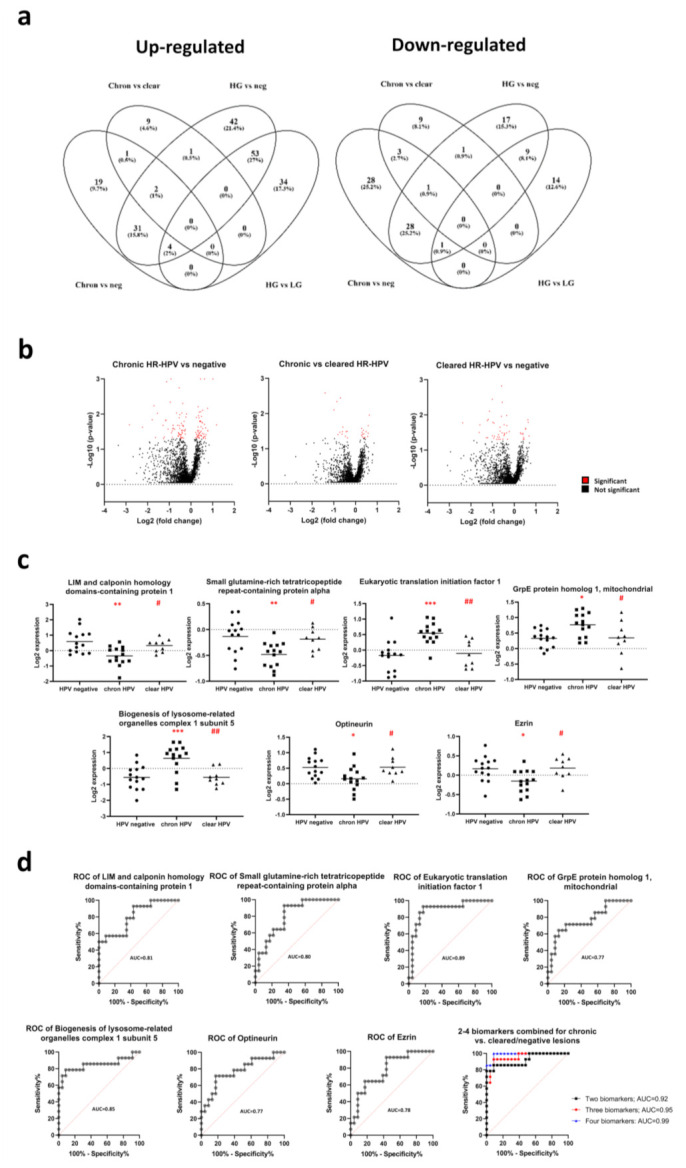
Venn diagram of up-regulated and down-regulated proteins between the chronic HR-HPV group, the cleared HR-HPV group, the high-grade lesion group, the low-grade lesion group and the negative group (**a**); volcano plots of log2 fold change vs. log10 *p*-value of changed proteins between the chronic HR-HPV group and the negative group; the chronic HR-HPV group and the cleared HR-HPV group; and the cleared HR-HPV group and the negative group, respectively. Red dots show significantly changed proteins and black dots show non-significantly changed proteins (**b**); the seven most changed proteins between the chronic HR-HPV group and the cleared HR-HPV and negative groups (**c**). * indicates a *p*-value of <0.05, ** indicates a *p*-value of <0.01 and *** indicates a *p*-value < 0.001 between the chronic HR-HPV group and the HR-HPV negative group. # indicates a *p*-value of <0.05 and ## indicates a *p*-value of <0.01 between the chronic HR-HPV group and cleared HR-HPV group. Receiver operating characteristic (ROC) curves distinguish a chronic HR-HPV sample from a cleared HR-HPV or negative sample. The area under the curve (AUC) is displayed within the graphs (**d**).

**Figure 4 cancers-13-05983-f004:**
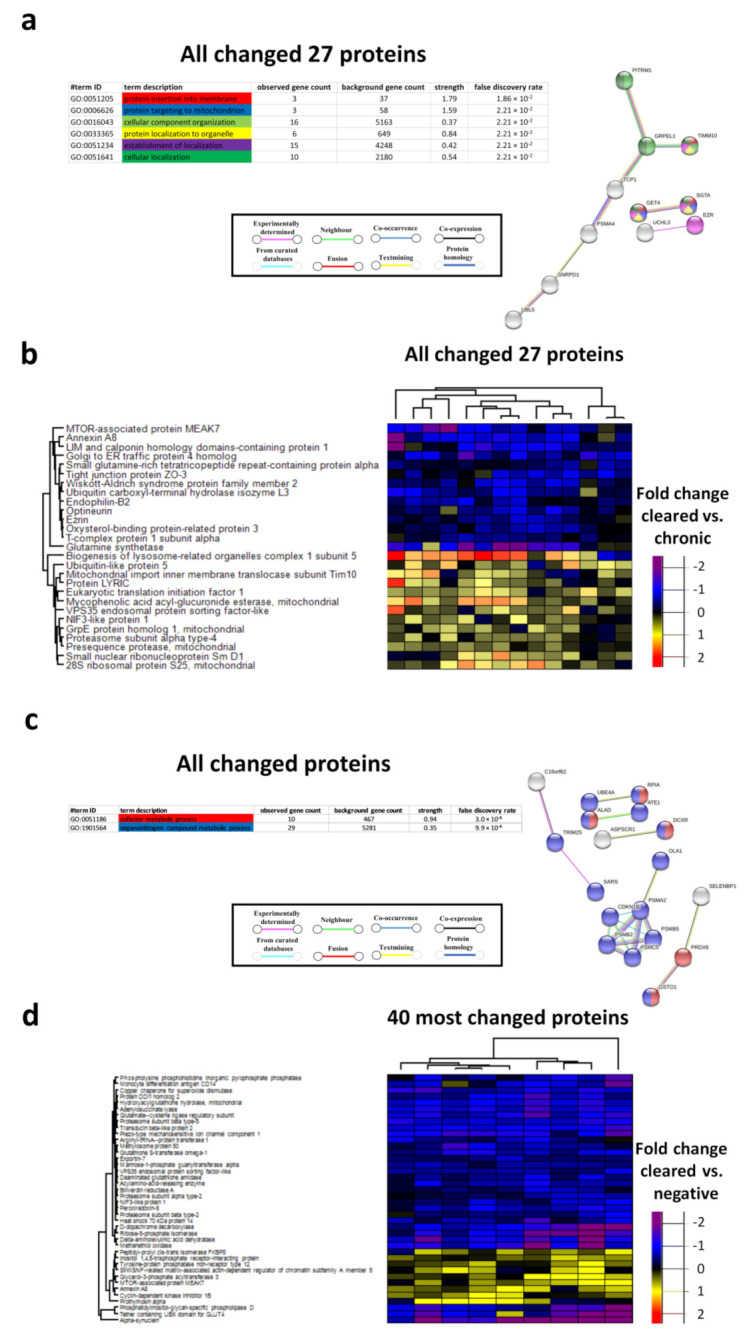
Protein-protein network analysis of the biological processes of the 27 changed proteins between the cleared HR-HPV group and the chronic HR-HPV group (**a**) and of all changed proteins between the cleared HR-HPV group and the HR-HPV negative group (**c**). The biological processes with the lowest FDRs are indicated by colour, and interactions are indicated by colour and line type; heatmaps of the 27 changed proteins between the cleared HR-HPV group and the chronic HR-HPV group (**b**) and of all changed proteins between the cleared HR-HPV group and the HR-HPV negative group (**d**). Disconnected nodes are hidden in the network.

**Figure 5 cancers-13-05983-f005:**
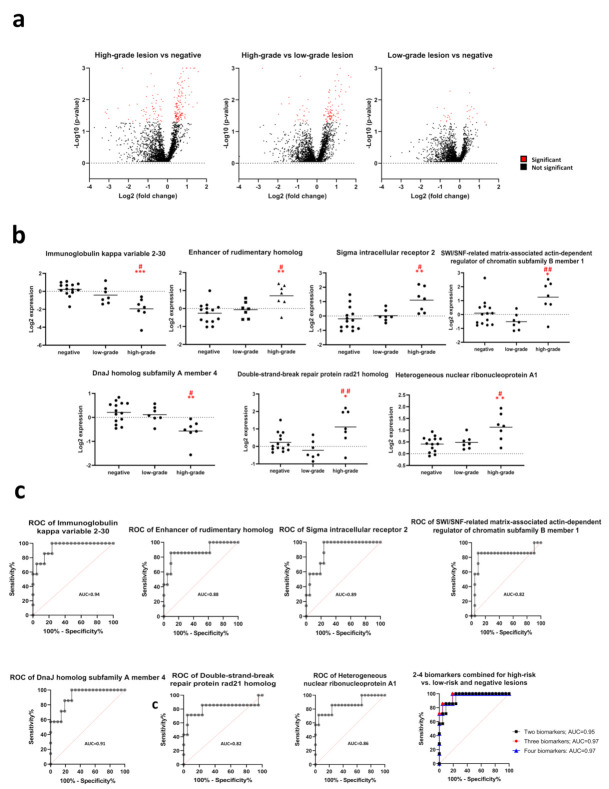
Volcano plots of log2 fold change vs. log10 *p*-value of changed proteins between the high-grade lesion group and the negative group; the high-grade HR-HPV group and the low-grade lesion group; and the low-grade lesion group and the negative group, respectively. Red dots show significantly changed proteins, and black dots show non-significantly changed proteins (**a**); the seven most changed proteins between the high-grade lesion group and the low-grade and negative groups (**b**). * indicates a *p*-value of <0.05, ** indicates a *p*-value of <0.01 and *** indicates a *p*-value < 0.001 between the high-grade lesion group and the negative group. # indicates a *p*-value of <0.05 and ## indicates a *p*-value of <0.01 between the high-grade lesion group and the low-grade lesion group. Receiver operating characteristic (ROC) curves distinguish a high-grade lesion from a low-grade or negative sample. The area under the curve (AUC) is displayed within the graphs (**c**).

**Figure 6 cancers-13-05983-f006:**
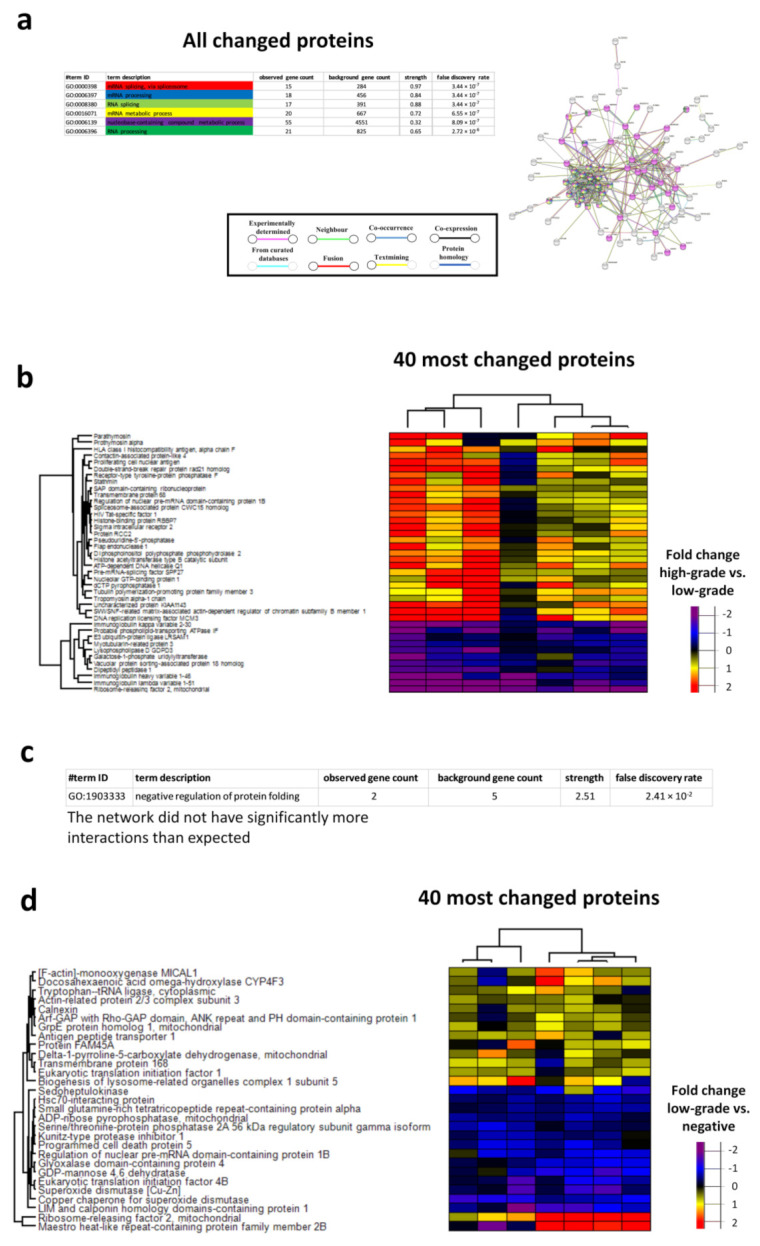
Protein-protein network analysis of the biological processes of all changed proteins between the high-grade lesion group and the low-grade lesion group (**a**) and between the low-grade lesion group and the negative group (**c**). The biological processes or molecular processes with the lowest FDRs are indicated by colour, and interactions are indicated by colour and line type; heatmaps of the 40 most changed proteins between the high-grade lesion group and the low-grade lesion group (**b**) and between the low-grade lesion group and the negative group (**d**). Immunoglobulin heavy variable 1–46, immunoglobulin lambda variable 1–51 and immunoglobulin kappa variable 2–30 could not be identified in the String database. Disconnected nodes are hidden in the network.

**Figure 7 cancers-13-05983-f007:**
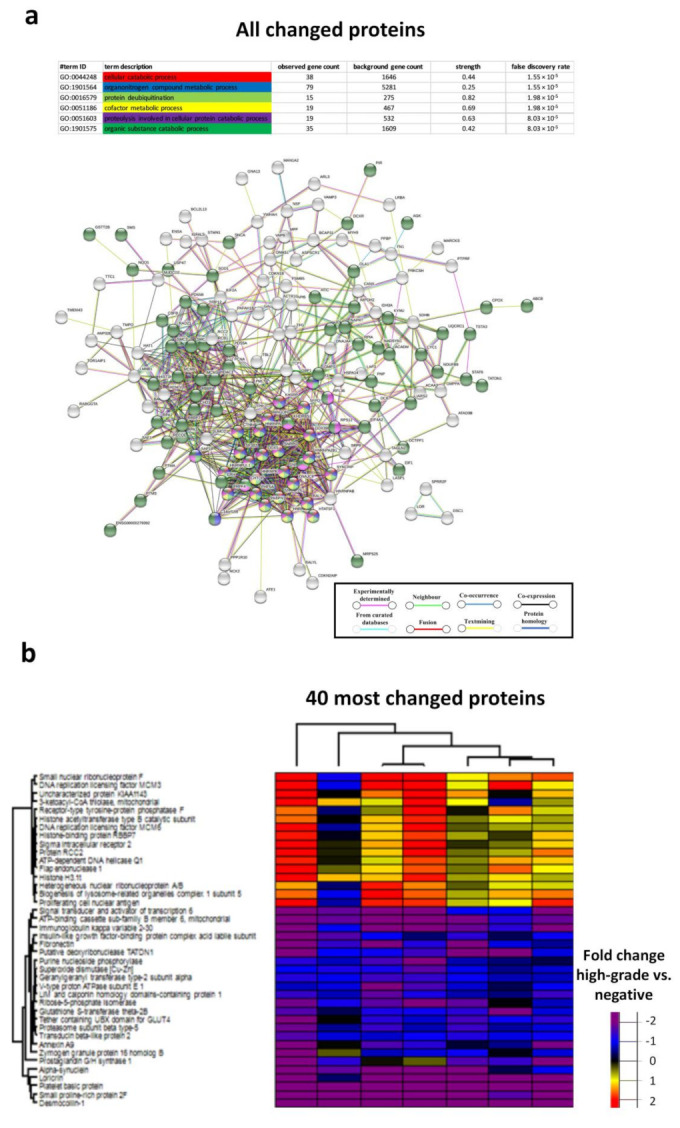
Protein-protein network analysis of the biological processes of all changed proteins between the high-grade lesion group and the negative group (**a**) and heat map of the 40 most changed proteins between the high-grade lesion group and the negative group (**b**). Immunoglobulin kappa variable 2–30 could not be identified in the String database. Disconnected nodes are hidden in the network.

**Table 1 cancers-13-05983-t001:** Age, HIV status, cytology and HPV types of the cohort. For the HPV chronic group, persistent HPV types are displayed, and in parenthesis are HPV types present at other sampling occasions.

Group A. HPV Incident					
	Age at Inclusion	HIV Status	Cytology Baseline	Cytology Follow-Up	HPV Types Follow-Up
Woman 1	45	Positive	Normal	Normal	66, 87
Woman 2	61	Positive	Normal	Not determined	52
Woman 3	40	Positive	Normal	Normal	45
Woman 4	60	Positive	Normal	Normal	68
Woman 5	48	Positive	ASCUS	Normal	45 (6)
Woman 6	47	Positive	Normal	Normal	31
Woman 7	31	Positive	Normal	Normal	16 (74)
**Group B. HPV Cleared**					
	**Age at Inclusion**	**HIV Status**	**Cytology Baseline**	**Cytology Follow-Up**	**HPV Types Baseline**
Woman 8	49	Positive	Normal	Normal	53, 91 (16)
Woman 9	55	Positive	Normal	Normal	58
Woman 10	45	Positive	Normal	Normal	16
Woman 11	44	Positive	Normal	ASCUS	56
Woman 12	50	Positive	Normal	Normal	69
Woman 13	45	Positive	HSIL	ASCUS	16
Woman 14	43	Negative	Normal	Normal	52
Woman 15	38	Negative	Normal	Normal	16
Woman 16	26	Negative	LSIL	Normal	51, 87
**Group C. HPV Chronic**					
	**Age at Inclusion**	**HIV Status**	**HPV Types**	**Cytology**	**Subgroup**
Woman 17	51	Positive	16	Normal	Low-grade
Woman 18	38	Positive	68, 70, 83, 86 (6, 31)	Normal	Low-grade
Woman 19	42	Positive	52, 53	ASCUS	Low-grade
Woman 20	48	Positive	52 (67, 90)	ASCUS	Low-grade
Woman 21	43	Positive	39, 52, 53	ASCUS	Low-grade
Woman 22	37	Positive	68 (87)	LSIL	Low-grade
Woman 23	36	Positive	70 (35)	LSIL	Low-grade
Woman 24	39	Positive	58	HSIL	High-grade
Woman 25	52	Positive	52, 58	HSIL	High-grade
Woman 26	53	Positive	31	HSIL	High-grade
Woman 27	47	Positive	31	HSIL	High-grade
Woman 28	43	Positive	16, 51, 52	HSIL	High-grade
Woman 29	43	Positive	33 (59)	AIS	High-grade
Woman 30	50	Positive	45, 70 (6, 11, 70, 89, 51)	SCC	High-grade
**Group D. Normal Cytology**					
	**Age at Inclusion**	**HIV Status**	**HPV Types**	**Cytology**	
Woman 31	41	Positive	Negative	Normal	
Woman 32	61	Positive	Negative	Normal	
Woman 33	56	Positive	Negative	Normal	
Woman 34	50	Positive	Negative	Normal	
Woman 35	45	Positive	Negative	Normal	
Woman 36	51	Positive	Negative	Normal	
Woman 37	39	Positive	Negative	Normal	
Woman 38	40	Positive	Negative	Normal	
Woman 39	40	Negative	Negative	Normal	
Woman 40	42	Negative	Negative	Normal	
Woman 41	55	Negative	Negative	Normal	
Woman 42	41	Negative	Negative	Normal	
Woman 43	33	Unknown	Negative	Normal	
Woman 44	40	Negative	Negative	Normal	

## Data Availability

The mass spectrometry proteomics data have been deposited to the ProteomeXchange Consortium via the PRIDE partner repository with the dataset identifier PXD029798 [47]. Additional data supporting reported results are available on request from the corresponding author.

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
