# Peer review of "Changes in the Proteome in the Development of Chronic Human Papillomavirus Infection—A Prospective Study in HIV Positive and HIV Negative Rwandan Women"

_cancers, 2021, doi:10.3390/cancers13235983_

Round 1

Reviewer 1 Report

Using cytology, HPV screening and proteomics, the authors aimed at identifying biomarkers that are characteristic for cleared or chronic HR-HPV infections. For this they analyzed cervical samples from Rwandan HIV+ and HIV- women at baseline, 9 months, 18 months and 24 months. Overall, this is an interesting study that may deserve publication after major revision.

Major comments

  • The illustrations are not presented at their best. The authors should strongly revise all illustrations. In doing so, they can take a look at many other similar publications.
  • Figure legends are missing!!
  • When printed the smaller images are not readable!
  • Is it known whether HIV proteins interact with or control the expression of the factors identified here?
  • Page 13: The authors may want to include in the conclusion section the identified biomarkers were found in HIV positive women with chronic HR-HPV infection and that it may be needed to further test the biomarkers in a bigger cohort of HIV negative patients to clarify their significance.

Author Response

We thank the reviewer for the valuable comments and suggestions.

Major comments

  • The illustrations are not presented at their best. The authors should strongly revise all illustrations. In doing so, they can take a look at many other similar publications. Figure legends are missing!!

Answer: We apologize for this mistake and added figure legends under figures. We have increased the size of the table so it is easier to read and slightly revised the figures.

  • When printed the smaller images are not readable!

Answer: In author instructions they recommended to incorporate figures in the text of the manuscript. Since the pictures contain a lot of data it is difficult to read out the text in the figures if the manuscript is printed. To facilitate reading of the figures we have now added all figures as tif-files with high resolution as supplementary materials.

  • Is it known whether HIV proteins interact with or control the expression of the factors identified here?

Answer: SGT may interact with Vpu and Gag of HIV-1 [1]. Ezrin may take part in HIV replication [2]. However, we do not think that HIV status is a bias for the biomarkers, since 76% of women in the control and cleared groups were HIV-positive in comparison with 100% in the chronic group. We have also analysed only HIV-positive cases showing that HIV status did not seem to constitute a bias for SGTA and ezrin.

  • Page 13: The authors may want to include in the conclusion section the identified biomarkers were found in HIV positive women with chronic HR-HPV infection and that it may be needed to further test the biomarkers in a bigger cohort of HIV negative patients to clarify their significance.

Answer: Since we only identified chronic HR-HPV infections among HIV+ women and 0% among HIV- women we are currently validating our findings in new HIV+ cohorts in Rwanda and Ethiopia. At present 50 women have been recruited in Kigali, Rwanda, and we plan to include 400 women in Rwanda and 400 women in Addis Ababa, Ethiopia. If we can confirm that biomarkers are valid in HIV+ women, we will assess the biomarkers also in HIV- women. However, since only few HIV- women develop chronic HR-HPV infections we would need to significantly increase the numbers of participants for a HIV- cohort. We have added information about our plans to validate in Conclusions.

Reviewer 2 Report

Dear Authors,

Thank you very much for this opportunity.
However the manuscript contains several flaw and as it stands it is hard to accept their conclusions.

At first, the result of cytology is not accurate.
The authors should show the final diagnosis using the colposcopy-guided biopsy samples.
Also the samples should be affected by the disease status of HIV and their past treatment.

Author Response

We thank the reviewer for the valuable comments and suggestions.

  • At first, the result of cytology is not accurate. The authors should show the final diagnosis using the colposcopy-guided biopsy samples.

Answer: As we state in Discussion we have not corroborated our findings from cytology with colposcopy or histology. The goal of the present study was, however, to identify biomarkers in cytology samples to screen out women at risk of developing cervical cancer without the use of resource demanding techniques such as colposcopy and histology for settings suitable in developing countries.

  • Also the samples should be affected by the disease status of HIV and their past treatment

Answer: We acknowledge that HIV status and antiretroviral treatment can affect the results. We have listed this as a potential bias in Discussion. Also we have added information on potential interference between HIV and biomarkers in Discussion.

Reviewer 3 Report

The manuscript titled “Changes in the Proteome in the Development of Chronic Human Papillomavirus Infection-A Prospective Study in HIV Positive and HIV Negative Rwandan Women” by  Bienvenu et al., is an interesting proteomic study performed with the aim to identify protein signature able to discriminate subgroups of patients. The study is scientifically sounded and the experimental workflow is well designed.

Unfortunatly, the experimental confirmation of the proposed biomarkers is missing. Authors should be verify  (LS-MS/MS? ELISA? Western blotting?) their model in an independent cohort of patients in order to make the paper suitable for publication.

Minor concerns:

  1. “HIV Positive and HIV Negative” in the title should be deleted, because this information is marginal in the text
  2. Figure legends are missing in the text. Please add
  3. Proteomic data should be deposited in a proteomic database, Proteome Exchange, for example
  4. The list of the identified proteins and quantification for all analysed groups and patients should be attached as supplementary data (excel data)
  5. Figure quality is low and figures are too small.
  6. The information’s regarding the created equations should be moved into MM section and only the best equation should be reported also in results.

Author Response

We thank the reviewer for the valuable comments and suggestions.

  • Unfortunatly, the experimental confirmation of the proposed biomarkers is missing. Authors should be verify (LS-MS/MS? ELISA? Western blotting?) their model in an independent cohort of patients in order to make the paper suitable for publication.

Answer: Cervical samples for proteomics are unfortunately used up and therefore we do not have the possibility to validate the findings with ELISA/western blot in this cohort. Since our definition of chronicity is a persistent infection during two years it will not be possible to validate the biomarkers for this study. However, we are currently conducting a follow-up study in Rwanda (HIV clinic, University Teaching Hospital, Kigali) and Ethiopia (HIV clinic, Black Lion Hospital, Addis Ababa) in HIV+ women where we will validate the current findings of biomarkers with ELISA and western blot. We have added this information in Discussion.

Minor concerns:

  • “HIV Positive and HIV Negative” in the title should be deleted, because this information is marginal in the text

Answer: We acknowledge that HIV is not the major topic of our study. However, since HIV is a risk factor for contracting HPV we think that it is informative to show that the study was performed in HIV+ and HIV- women.  

  • Figure legends are missing in the text. Please add

Answer: We apologize for this mistake and have added figure legends in the manuscript.

  • Proteomic data should be deposited in a proteomic database, Proteome Exchange, for example

Answer: We thank the reviewer for the suggestion. We have uploaded the data in Proteome Exchange.

  • The list of the identified proteins and quantification for all analysed groups and patients should be attached as supplementary data (excel data)

Answer: We have attached as supplemental data.

  • Figure quality is low and figures are too small.

Answer: In author instructions they recommended to incorporate figures in the text of the manuscript. We had significant difficulties uploading figures with higher resolution. We have therefore now added all figures as tif-files with high resolution as supplemental materials.

  • The information’s regarding the created equations should be moved into MM section and only the best equation should be reported also in results.

Answer: We have moved the information about how equations was created under Methods.  

Round 2

Reviewer 1 Report

As far as the answers to my questions are concerned, I am also satisfied with the revised manuscript. 

Author Response

We thank Reviewer 1 for valuable suggestions and comments.

Reviewer 2 Report

They have replied to my concerns. 

Author Response

(The authors gave the same response as above.)

Reviewer 3 Report

Authors addressed the reviewer suggestions. Only the figure format should be improved. 

Author Response

We thank Reviewer 1 for valuable suggestions and comments. We have revised the figures so they now are readable and not compressed in Word as they were in previous versions of the manuscript.